# An extensive investigation of the ability of the ICOLMDZ model to simulate a katabatic wind event in Antarctica.

Valentin Wiener<sup>1</sup>, Étienne Vignon<sup>1</sup>, Thomas Caton Harrison<sup>2</sup>, Christophe Genthon<sup>1</sup>, Felipe Toledo<sup>3</sup>, Guylaine Canut-Rocafort<sup>4</sup>, Yann Meurdesoif<sup>5</sup>, and Alexis Berne<sup>6</sup>

Abstract. Katabatic winds are a key feature of the climate of Antarctica, but despite decades of extensive studies, substantial biases remain in their representation in atmospheric models. However, it is often difficult to identify the origin of those biases amongst the model resolution, physical content, and large-scale forcings aspects. This study conducts an extensive investigation of the ability of the ICOLMDZ atmospheric model to simulate Antarctic katabatic winds by disentangling uncertainties associated with parameter calibration, from those associated with horizontal resolution as well as structural deficiencies in the model with a particular attention given to turbulent diffusion. We carefully select a katabatic-driven wind event in clearsky conditions in Adélie Land, and perform perturbed parameter experiments at three different horizontal resolutions (10, 20 and 40 km). ICOLMDZ is able to reproduce wind observations, but the parametric uncertainty remains large and structural differences not associated with parameter calibration nor horizontal resolution are found for turbulence and near-surface temperature. A parametric analysis reveals that the most critical parameter controlling the magnitude of near-surface winds is roughness length, whereas near-surface temperatures are mainly controlled by snow near-infrared albedo. Sensitivity to horizontal resolution reveals that the 40-km configuration compares least favourably with the observations, and that the 10-km and 20-km ensembles cannot be distinguished due to a too wide parametric uncertainty. We then discuss three aspects of katabatic winds modeling that we deem critical but underappreciated: the parameterization of roughness length over snow, oscillations in katabatic flows, and the representation of subgrid-scale orographic drag. This study underlines in particular the need for a more physical parameterization of roughness length to correctly represent near-surface wind along the slopes of Antarctica.

<sup>&</sup>lt;sup>1</sup>Laboratoire de Météorologie Dynamique, Institut Pierre-Simon Laplace, Sorbonne Université/CNRS/École Polytechnique – IPP, Paris, France

<sup>&</sup>lt;sup>2</sup>British Antarctic Survey, Cambridge, United Kingdom

<sup>&</sup>lt;sup>3</sup>Laboratoire Atmosphère, Milieux et Observations Spatiales, Institut Pierre-Simon Laplace, UVSQ Université Paris-Saclay, Sorbonne Université, CNRS, Guyancourt, France

<sup>&</sup>lt;sup>4</sup>CNRM-Université de Toulouse, Météo-France/CNRS, 42 ave. G. Coriolis, 31057, Toulouse, France

<sup>&</sup>lt;sup>5</sup>Laboratoire des Sciences du Climat et de l'Environnement, Institut Pierre-Simon Laplace, CEA/CNRS/UVSQ, Gif-sur-Yvette, France

<sup>&</sup>lt;sup>6</sup>Environmental Remote Sensing Laboratory, Swiss Federal Institute of Technology in Lausanne, Lausanne, Switzerland **Correspondence:** Étienne Vignon (etienne.vignon@lmd.ipsl.fr)

## 1 Introduction

The strongest average near-surface winds on Earth are found on the Antarctic continent. These winds are widely known as katabatic winds, and as historically described by e.g. Parish and Bromwich (1987), originate from the strong cooling of the air just above the slopes of the Antarctic plateau, which is particularly intense in winter during polar night. This near-surface air, colder and therefore denser than the air at the same elevation but further downslope, flows down the slopes of Antarctica towards the coast, while being accelerated by gravity and deflected by Coriolis force. As they are strongly influenced by topography, these winds exhibit a high directional constancy, even in summer due to the adjustment of the large-scale ambient pressure gradient to the continental orography (Parish and Cassano, 2001), and may exceed 15 m s<sup>-1</sup> in yearly average in certain regions such as Adélie Land (Parish, 1988). In summer, near-surface winds show a diurnal cycle due to variations of surface absorbed solar radiation (Heinemann, 1997).

Antarctic near-surface winds are also strongly influenced by synoptic pressure gradients such as variations of the Southern Hemisphere circumpolar vortex (Van den Broeke and Van Lipzig, 2002), gradients that are often parallel to the katabatic forcing and in average directed from the inner plateau to the coast (Van den Broeke and Van Lipzig, 2003). The designation "katabatic winds" used in this study is therefore somewhat simplified as katabatic forcing is not the only driver of Antarctic near-surface winds.

The Antarctic katabatic flow presents a wind speed maximum located in the first few hundred meters above ground level and generally between 10 to 20 m s $^{-1}$  (Heinemann and Zentek, 2021), capped by a temperature inversion (e.g., Guest (2021)). Katabatic winds are also characterized by an intense turbulent activity due to a strong vertical wind shear that impacts in particular surface heat and momentum exchanges.

Moreover, sudden lulls in the intensity of katabatic winds may be observed at the coast associated with pressure drops (Ball, 1956) and an abrupt thickening of the boundary layer. From supercritical (Froude number Fr > 1), the flow turns subcritical (Fr < 1) (Yu et al., 2007). This phenomenon, which often occurs along the coast of Antarctica, is hereafter referred to as a katabatic jump (e.g., Gallée and Pettré (1998); Vignon et al. (2020)).

40

Antarctic katabatic winds have a critical role in the climate system at the continental scale but also at global scale. By their force, they generate blowing snow that can be sublimated over the polar cap or even carried off the continent, thus contributing negatively to the Antarctic surface mass balance (SMB) (Gerber et al., 2023). Moreover, the katabatic flow, dry and adiabatically warmed during its descent, can also make precipitation sublimate before reaching the surface (Grazioli et al., 2017; Jullien et al., 2019), a phenomenon not to be neglected in Antarctic coastal margins where both wind and precipitation are significant. The relative importance of the processes listed above on the SMB are discussed in Agosta et al. (2019). Katabatic winds also play a key role at the coast by dislocating and pushing sea ice offshore: the ocean ice-freed areas called polynyas cool at the contact of the colder air and freeze again, increasing the density of the water underneath and contributing to the formation of Antarctic bottom water (Abernathey et al., 2016) which stores heat and carbon over centuries, hence impacting global climate. Conversely, these polynyas can alter near-surface wind through the induced extra land breeze circulation and destabilization of

However, substantial biases in the representation of katabatic winds are still reported even for state-of-the-art polar climate models. Davrinche et al. (2024) find relatively low correspondence (determination coefficients between 0.5 and 0.7) between the MAR (Modèle Atmosphérique Régional) model near-surface winds and the observations in Adélie Land at a 3-hourly frequency, and particularly in winter at the coast. Van Wessem et al. (2014) find an even lower determination coefficient (0.28) on average between the RACMO model and automatic weather stations (AWS) in Dronning Maud Land. Souverijns et al. (2019) report a general overestimation of the COSMO-CLM<sup>2</sup> model near-surface winds over the Antarctic ice sheet of 2 to 5 m s<sup>-1</sup>. Sanz Rodrigo et al. (2013) report wind speed biases up to 10 m s<sup>-1</sup> locally in the ERA-Interim reanalysis, as well as a general underestimation of the wind speed variability. Vignon et al. (2018) find large biases between the LMDZ general circulation model and the AWS in the slopes of Adélie Land, with wind speed up to twice too weak. Gossart et al. (2019) report for three reanalysis products an underestimation of the mean wind speed in the interior of the continent, even more significant at the coast where the annual mean bias reaches approximately 25%. It is also found that the strongest wind speeds are not reproduced by all the reanalysis products. Caton Harrison et al. (2022) compare reanalyses with scatterometer observations for near-surface winds and with radiosondes on the vertical, and report substantial wind speed biases in coastal margin areas. They also find an underestimation of high near-surface wind speed associated with strong synoptic forcing, and that the variability of more katabatic-driven winds is not well represented. Vignon et al. (2019) find significant differences in the wind profiles between reanalyses and radiosonde observations at the coast, and perform a free simulation of the Polar-WRF regional model that exhibits too strong and too shallow katabatic layers over the entire continent.

70

This non-exhaustive list of biases originates from various modeling challenges in the representation of Antarctic winds. Davrinche et al. (2024) show that unraveling the contribution of large-scale drivers from local drivers is crucial to decipher the origin of Antarctic wind biases, in particular for strongest winds speeds when the synoptic forcing and the katabatic forcing are aligned.

Furthermore, model winds are highly sensitive to both horizontal and vertical resolution. Reanalyses with typical spatial resolution of 0.25° or lower are unable to represent complex topographic issues such as channeling in escarpment areas (Sanz Rodrigo et al., 2013). In general, lower spatial resolutions induce a too smooth wind variability compared to observations (Gossart et al., 2019). Lenaerts et al. (2012) assess with a regional climate model that enhancing horizontal resolution (from 27 to 5.5 km) strongly reduces wind speed biases in Adélie Land. Processes related to the complex topography at the coast such as katabatic jumps are also more difficult to represent in coarser resolution models (Vignon et al., 2019). Caton Harrison et al. (2022) show that near steep coastal slopes, reanalyses wind speed biases are larger for locally-driven events than for events in which synoptic forcing is dominant. Concerning vertical resolution, Heinemann and Zentek (2021) show that models with a too wide vertical spacing are not able to reproduce very shallow katabatic layers. It is stated that 10 vertical levels below 300 m are sufficient to resolve the katabatic layer over the Antarctic slopes.

Another important aspect in the modeling of Antarctic winds is the representation of turbulent mixing in the katabatic layer.

Indeed, Van Der Avoird and Duynkerke (1999) show that the wind speed maximum in altitude is dependent on turbulent mixing, and impacts in turn the generation and dissipation of the turbulent kinetic energy (TKE). Turbulence is also crucial for the mixing of momentum at the top of the katabatic layer and for explaining the deepening thereof associated with katabatic jumps (Yu and Cai, 2006). The parameterization of subgrid-scale orographic drag and subgrid-scale orography parameters such as the standard deviation of high-resolution topography are also critical (Pithan et al., 2015).

More indirect effects also need to be considered. Bromwich et al. (2001), who also stress the importance of large-scale forcing and turbulence parameterization for the representation of katabatic flows over the slopes of the Greenland ice sheet, further underline a sensitivity of near-surface winds to cloud cover and cloud radiative properties that impact near-surface temperatures, and in turn the intensity of katabatic winds.





The aim of this study is to extensively assess the representation of katabatic winds in the ICOLMDZ atmospheric model. To achieve this, we disentangle three aspects which contribute to uncertainty in their representation: parameter calibration, horizontal resolution, and structural deficiencies in the physical parameterizations of the model. We focus on a clear-sky katabatic wind event that occurred in the Adélie Land region, which has the benefit of being instrumented while experiencing intense and frequent katabatic winds. The wind event is mostly driven by katabatic forcing with limited effects of large-scale forcing by synoptic conditions. Simulations are run with a perturbed parameter ensemble (PPE) framework to perform an exhaustive generalized sensitivity study with no a priori on the tuning parameter value within their uncertainty range. Particular attention is given to turbulence thanks to new sonic anemometer observations collected in coastal Adélie Land.

The paper is structured as follows: the observation datasets are described in Sect. 2.1. The choice and description of the case study is presented in Sect. 2.2, before introducing the ICOLMDZ model in Sect. 2.3, and the simulation configuration as well as the analysis methods in Sect. 2.4. A first PPE is presented in Sect. 3.1 and its sensitivity to physics parameterizations is explored in Sect. 3.2. Sensitivity to horizontal resolution is investigated in Sect. 3.3. Further discussions on issues specific to the modeling of Antarctic near-surface winds are proposed in Sect. 4, before a conclusion with perspectives in Sect. 5.

# 2 Data, Model and Methods

## 110 2.1 Observation Datasets for Model Evaluation

Three observation sites named D18, D47 and D85, are selected to evaluate the ICOLMDZ model along the Adélie Land transect (see Fig. 4). These sites are located on the logistical track from the coastal station Dumont d'Urville (DDU) to Concordia station at Dome C on the Plateau, along which katabatic winds develop and strengthen. The slope is steeper near the coast, reaching an elevation of 1560 m at D47 less than 100 km inland, and becomes flatter as we approach the Plateau. The longitudes, latitudes and altitudes of D18, D47 and D85 are respectively [139.69; -66.73; 468 m]; [138.72; -67.39; 1560 m] and [134.15; -70.43; 2650 m]. We do not consider the Concordia station as it is located upon a dome, and consequently experiments no local katabatic winds. In the following sections, we describe the observation datasets available at those sites.

## 2.1.1 The D18-AWACA mast








During the austral summer season 2022-2023, a 7-m meteorological mast was deployed at the D18 site, approximately 16 km south-west of the Dumont d'Urville French permanent station in a region of high near-surface winds (roughly  $10 \text{ m s}^{-1}$  at 2 m on average according to Davrinche et al. (2024)). This mast pictured in Fig. 1 has been deployed just one kilometer further upslope than the already instrumented D17 site (see Barral et al. (2014)), which is very useful to cross-validate data. Terrain at D18 is slightly sloped (on average,  $1.8^{\circ}$  between D17 and D18) and entirely covered by snow (neither blue ice nor outcropping rocks) forming sastrugi under the influence of wind. As there is no other obstacle, the roughness of the surface is mainly due to the sastrugi. In general, snow-covered surfaces are characterized by small roughness lengths but that may spread over very variable scales. For example, Vignon et al. (2017a) find roughness length values between  $10^{-5}$  and  $10^{-2}$  m at Concordia.

The conception and deployment of this mast took place in the context of the AWACA project (https://awaca.ipsl.fr/, last access 16/01/2025), which aims at understanding the origin, transformation and deposition of atmospheric water in Antarctica.

In addition to classical meteorological variables such as wind speed (A100LK cup anemometer) and direction (W200P windvane), pressure (BaroVue10 barometer), temperature and humidity (HMP155 thermohygrometer), this mast includes 2 drifting snow sensors (FlowCapt<sup>TM</sup> FC4) measuring blowing snow horizontal mass fluxes respectively in the first and second meter above the surface; a radiometer measuring downwelling and upwelling shortwave and longwave fluxes (SN500SS); and turbulence variables from a sonic anemometer (METEK uSonic3-Omni) that samples high-frequency (10 Hz) 3D-wind and temperature upstream of the south-east mean wind. The instruments height above the snow surface is monitored by an acoustic snow-height gauge (SR50A). Turbulence variables such as turbulent kinetic energy (TKE), friction velocity ( $u_*$ ) and sensible heat flux (H) are estimated from raw sonic anemometer outputs using an eddy covariance data processing algorithm designed in the framework of the AWACA project, and validated with the Météo-France AIDA processing. The sampling frequency is 10 Hz, and the integration time is 30 minutes as recommended in Aubinet et al. (2012). Ogive tests have been performed to ensure that no energy domain is cut off with this integration time. Classical meteorological variables are collected with a timestep of 1 minute, but are later averaged over 10 minutes to match the AWS timestep (see Sect. 2.1.2).

According to the manufacturer, the uncertainty of the cup anemometers is 1% + 0.1 m s<sup>-1</sup>;  $2^{\circ}$  for the wind vane, where we add  $3^{\circ}$  to take into account the uncertainty in the orientation initial setting; and  $\pm$  (0.226 - 0.0028 T) °C for temperature. Uncertainties of the sonic anemometer come from the stochastic nature of the measured turbulence, errors in the instrument measurements, and variability of the footprint sampled in heterogeneous environments, the latter being probably negligible due to the smoothness of the surface considered. The total random uncertainty ranges between 10 and 20% (Finkelstein and Sims, 2001). In the specific Antarctic context, additional uncertainties arise such as blowing snow (Sigmund et al., 2022) that may block ultrasounds pathways, or vibrations of the mounting arm caused by strong winds (Gao et al., 2024). The impact of blowing snow on the estimation of turbulence variables is discussed in Sect. 3.1.

The acoustic snow-height gauge indicated a snow accumulation of approximately 30 cm ( $\pm$  10 cm) at the time of the case study. Therefore, the sonic anemometer height is approximately 2 m, and the upper wind level around 6.7 m ( $\pm$  10 cm).

**Figure 1.** The AWACA-D18 meteorological mast, scheme (left panel) and photograph taken on the 14th of January 2023, six days before the case study. Note that the intermediate level of wind, humidity and temperature measurements at 4.5 m has been installed the following season.

# 2.1.2 The AMRDC-AWS

The Antarctic Meteorological Research and Data Center Automatic Weather Stations (AMRDC-AWS) is a network of self-powered meteorological stations deployed over all Antarctica, although rather scarcely on the inner eastern plateau, coordinated by the Antarctic Meteorology Project of the Madison College, Wisconsin. Stations at the D47 and D85 sites contain a Young 05103 aerovane measuring wind speed and direction, a thermo-hygrometer and a pressure sensor, but no snow-height gauge. Instrument height is estimated to be approximately 3 meters during the time of our case study (personal communication of D. Mikolajczyk), although it is difficult to evaluate the uncertainty of this estimation. Quality-controlled 10-min data have been

retrieved on the AMRDC Data Repository website (AMRDC, 2023). Uncertainties for the wind data have been taken from Wang et al. (2023):  $0.3 \text{ m s}^{-1}$  for wind speed and  $3^{\circ}$  for wind direction.

# 160 2.2 Choice and Description of the case study




Antarctic near-surface winds are complex as they depend on several factors such as the local katabatic effect, the large-scale pressure gradient, as well as cloud cover and blowing snow particles that can both affect the surface energy balance and the local buoyancy with possible feedback on the katabatic forcing and effects on the structure of the flow. To carry out an evaluation of the model on clear-sky katabatic-driven winds with a specific focus on the performance of the turbulent mixing and surface-layer parameterizations, we have to carefully select a case study with no clouds, as little blowing snow as possible, and with a limited contribution of synoptic forcing to reduce the uncertainty associated with lateral boundary conditions prescribed to the model and maximize the role played by the model within the inner domain. Therefore, we define below a list of criteria to select an event that will allow us to focus on these particular aspects of Antarctic winds:

- a representative katabatic flow, whose main forcing is gravity driven rather than synoptic large-scale conditions. We ascertain the so-called "katabaticness" of the event through the momentum budget approach from Caton Harrison et al. (2024) inspired from Van den Broeke and Van Lipzig (2003) with ERA5 reanalyses, that enables us to separate large-scale forcing from local forcings.
  - clear-sky conditions, to be free of clouds and precipitation impacts on the boundary layer that may have indirect effect on the wind. Clouds near the coast are monitored by a Vaisala CL31 ceilometer deployed at the DDU station.
- no gaps in the data. Various experimental issues such as glaze ice, a corrupted SD-card, defective sensors and wind-ripped solar panels frequently interrupted data collection of the D18 mast, drastically reducing eligible days particularly in winter. The AMRDC-AWS also presents gaps in the data that have an impact on the final selection.
  - little drifting snow, as it may alter near-surface thermodynamics and turbulence (Gallée et al., 2001; Bintanja, 2001), and because this process is not parameterized in the model yet. Moreover, drifting snow may block the ultrasound pathways of the sonic anemometer and contaminate the data.

After careful investigation, the local night of the 20th of January, 2023 is chosen as a good candidate for our case study. Fig. 2.a shows the wind speed evolution at D18. A strong wind event strengthens almost linearly from local afternoon (as a reminder, local time LT is UTC +10) throughout the night to reach a peak around 07:00 LT at 17.8 m s<sup>-1</sup> at 7 meters in the following local morning, before decreasing rapidly. The wind shear between 2 and 7 m follows the same evolution and simultaneously reaches a maximum of 0.6 m s<sup>-1</sup> m<sup>-1</sup> (not shown). Wind at D47 (Fig. 2.c) shows the same peak although without the linear increase phase and a relatively flatter shape. Conversely, no clear peak can be identified at any time at D85 as the slope is less steep. This hints that the katabatic event is limited to coastal Adélie Land and not generated by synoptic-scale forcing. From now on, the study focuses only on D18 and D47.

Wind direction at D18 and D47 (Fig. 2.b,d) goes from south-south-east to south, indicating wind coming from the Plateau. No

**Figure 2.** 10-minute wind speed (left) and direction (right) at the D18 (top), D47 (middle) and D85 (bottom) sites for the 20-21 January 2023, with instrumental uncertainties in gray shading. Wind is measured at 6.7 m for D18 and approximately 3 m for D47 and D85. Wind speed measurements at 2 m at D18 are also plotted in black dotted line in the upper left graph.

clouds are reported by the DDU ceilometer (not shown), and wind data are available for the 3 sites. Incidentally, we choose not to use the DDU radiosondes to investigate the katabatic flow on the vertical, as they are launched near 10:00 LT (i.e. 00:00 UTC), 3 hours after the wind maximum when the wind has already largely subsided. Moreover, DDU is located upon an island 5 km off the coast and complex topographic issues arise that may complicate the interpretations.

However, a slight amount of drifting snow (below 5 g m  $^{-2}$  s  $^{-1}$ ) is reported by the uppermost AWACA-D18 FlowCapt (between

1 m and 2 m). This is an issue we cannot brush off lightly, as the upper drifting snow sensor measurements are clearly correlated with the number of gaps in the sonic anemometer data (not shown). As we do not see spikes in the data but rather gaps, we apply a threshold of 1% NaNs (Not a Number) for the sonic anemometer indicating a lower degree of confidence in the data. The lowermost FlowCapt™ measured substantially larger drifting snow (peaks above 100 g m<sup>-2</sup> s<sup>-1</sup>), but as the instrument is located well below the sonic anemometer height we consider that blowing snow particles did not block the sonic anemometer ultrasounds pathways and had a limited impact on turbulence data, although the indirect effect of drifting snow sublimation on temperature and humidity cannot be quantified.

The "katabaticness" of the 20th of January, 2023 (UTC) has been confirmed by the momentum budget decomposition following the methodology of Caton Harrison et al. (2024). Fig. 3 shows that the contribution of katabatic forcing predominates over large-scale forcing in the coastal area near D18 and D47.

**Figure 3.** Katabatic (a) and large-scale (b) daily-averaged, as a percentage of all "active" acceleration terms (see Sect. 3d of Van den Broeke and Van Lipzig (2003)) on 20th January 2023 (UTC). Blue crosses show the positions of D18, D47 and D85 in that order from the coast.

At D47, the daily-averaged katabatic forcing (term KAT in Eq. 2 of Van den Broeke and Van Lipzig (2003)) is 9.1 m s<sup>-1</sup> h<sup>-1</sup> whereas large-scale forcing is 3.9 m s<sup>-1</sup> h<sup>-1</sup>; and at D18 katabatic forcing is 16.4 m s<sup>-1</sup> h<sup>-1</sup> whereas large-scale forcing is 4.9 m s<sup>-1</sup> h<sup>-1</sup>. Therefore, the wind and its structure will be mainly controlled by the model physics and local forcing rather than large-scale effects.

## 2.3 The ICOLMDZ model

The ICOLMDZ model comprises the hydrostatic dynamical core DYNAMICO (Dubos et al., 2015) coupled to the physics of the atmospheric general circulation model LMDZ (Hourdin et al., 2020). This coupling, already used for instance in Bourdin

et al. (2024) remains quite recent and offers the advantage of running LMDZ with an icosahedral mesh, especially useful at the poles where classical longitude-latitude mesh narrows due to the convergence of meridians towards a singularity.

The LMDZ model is the atmospheric component of the French Institut Pierre-Simon Laplace (IPSL) global climate model, that participated in the CMIP exercises since CMIP-4 and that is currently preparing for CMIP7. While the Antarctic climate simulated by previous versions of LMDZ was already investigated almost thirty years ago (Krinner et al., 1997), recent studies dig into the representation of the stable boundary layer over the Plateau (Vignon et al., 2018), and clouds and precipitation evaluation over the DDU station (Lemonnier et al., 2021; Roussel et al., 2023). All these studies of the LMDZ Antarctic climate used the regular longitude-latitude mesh.



In this study, we use the CMIP6 version of the physics (hereafter called LMDZ-6A), thoroughly described in Hourdin et al. (2020) with further details in Madeleine et al. (2020) concerning the cloud parameterization and in Cheruy et al. (2020) for the surface-atmosphere coupling. However, we make a notable exception with respect to LMDZ-6A by using a new TKE-l turbulent diffusion scheme presented in Vignon et al. (2024), which has been specifically developed to improve the representation of stable boundary layers and has been designed to be fully tunable with as few parameters as possible, which will be critical in the study of katabatic winds. Furthermore, the model comprises a parameterization of the drag due to subgrid-scale orography (Lott and Miller, 1997; Lott, 1999). The intensity of the drag on the flow is characterized by subgrid-scale orography parameters derived from high-resolution topography. These parameters depend on the horizontal resolution of the model. This parameterization is deactivated in our simulation for reasons discussed in Sect. 4.

In the turbulent diffusion scheme of Vignon et al. (2024), the eddy diffusion coefficient for momentum  $K_m$  in near stationary conditions ( $\partial_t$  TKE  $\approx$  0) can be given by :

$$K_m = l^2 \sqrt{S^2} F_m(Ri) \tag{1}$$

where l is the mixing length,  $S^2$  the vertical wind shear  $(\partial_z u)^2 + (\partial_z v)^2$ , and  $F_m(Ri)$  the stability function depending on the Richardson number Ri.

The turbulent mixing length l is a combination of a ground-limited vertical length scale in neutral conditions and a stratification dependent length scale for stable conditions  $l_s$ .  $l_s$  can depend only on the stratification of the flow (Eq. 2), or on both stratification and wind shear (Eq. 3). The latter option has not yet been tested for strong wind shear conditions such as those encountered in a katabatic wind event. By default, the formulation using both wind shear and stratification (Eq. 3) is used.

$$l_s = c_l \frac{\sqrt{e}}{N} \tag{2}$$

$$l_s = c_l \frac{\sqrt{e}}{2\sqrt{S^2}(1+\sqrt{Ri}/2)}$$
 (3)

 $c_l$  is a coefficient ranging from 0.1 to 2, e is the TKE, and N the Brunt-Väisälä frequency. Thus, the value of the mixing length in stable boundary layers such as in katabatic flows is controlled by the parameter  $c_l$ .

The stability function  $F_m(Ri)$  is given by :



$$F_m(Ri) = S_m^{3/2} \sqrt{c_\epsilon (1 - \frac{Ri}{Pr})} \tag{4}$$

$S_m$  is a stability function depending on Ri,  $c_\epsilon$  a parameter controlling the dissipation rate  $\epsilon = \frac{e^{3/2}}{c_\epsilon l}$  which ranges between 1.2 and 10, and Pr the turbulent Prandtl number. Thus,  $\sqrt{c_\epsilon}$  modulates the amplitude of the diffusion coefficient for momentum. For more details about this parameterization, we refer the reader to Vignon et al. (2024).

In LMDZ, surface turbulent fluxes are computed using the Monin-Obukhov similarity theory with a bulk Richardson number  $Ri_b$  as a stability parameter. Momentum flux  $\tau$  and sensible heat flux H read:

$$\tau = \frac{\kappa^2}{\ln(z/z_{0m})^2} f_m(Ri_b) U^2 \tag{5}$$

$$H = -\rho c_p \frac{\kappa^2}{\ln(z/z_{0m})\ln(z/z_{0h})} f_h(Ri_b) U(\theta_v - \theta_{vs})$$
(6)

where  $\kappa$  is the von Karman constant ( $\kappa = 0.4$ ),  $z_{0m}$  and  $z_{0h}$  are roughness lengths for momentum and heat,  $f_m(Ri_b)$  and  $f_h(Ri_b)$  are stability functions depending on the bulk Richardson number, U is the wind speed at z, and  $\theta_v$  and  $\theta_{vs}$  are the virtual potential temperatures respectively at z and at  $z_{0h}$ . In the model,  $z_{0m}$  and  $z_{0h}$  are by default both set constant to  $10^{-3}$  m over ice caps. Note that in blowing snow conditions often provoked by kabatic winds, Sigmund et al. (2022) show that the Monin-Obukhov formulation of the turbulent fluxes becomes inadequate and must be revisited.

Diffusion of heat in the snow cover is parameterized as a conductive process with a constant thermal inertia  $I = \sqrt{\lambda_s \rho_s C_s}$ , with  $\lambda_s$  the snow thermal conductivity,  $\rho_s$  the snow density and  $C_s$  the snow specific heat per unit mass. 11 e-folding layers are used to cover timescales ranging from 1800 s to 240 y. Snow thermal inertia is fixed to 350 J m<sup>-2</sup> K<sup>-1</sup> s<sup>-1/2</sup> over polar caps.

Snow albedo Ab over ice caps is also fixed in LMDZ, although separated between the visible domain ( $Ab_{VIS} = 0.96$ ) and the near-infrared domain ( $Ab_{NIR} = 0.68$ ). The values of the snow thermal inertia and albedo are important as they control the amplitude of the temperature diurnal cycle, that may have indirect effects on near-surface winds.

The parameterization of shallow convection does not activate over the period and region of interest; nor the cloud scheme due to clear-sky conditions during the event.

# 2.4 Model Configuration and Experimental Setup







A specific configuration of ICOLMDZ is set up to properly model the chosen katabatic wind event. A limited area (LAM) version of DYNAMICO has been recently developed that enables regional studies at higher resolution and at reasonable computation cost, which will prove particularly critical at the coast where issues related to complex topography take place. We refer the reader to Sect. 2 of Raillard et al. (2024) which fully describes the ICOLMDZ LAM version. The LAM is forced by ERA5 reanalyses described in Hersbach et al. (2020) at the lateral boundaries, and ERA5 daily mean sea ice and sea surface temperature are prescribed. On the 20th of January 2023, near the end of the austral summer, ERA5 indicates that sea ice has melted or drifted away off the coast of Dumont d'Urville.

In order to constrain as much as possible large-scale meteorology, the wind is nudged to ERA5 above the boundary layer (above the hybrid  $\sigma$ -pressure level corresponding to a pressure of  $\approx$  750 hPa above sea level) with a relaxation time of 1 hour. With such a nudging, the synoptic meteorology is tightly constrained, but the low atmosphere including the katabatic layer is let free. It is worth mentioning that Caton Harrison et al. (2022) show that ERA5 are the best reanalyses available for Antarctica in terms of wind speed, which makes us confident regarding the representation of the large-scale conditions.

The simulation starts on 18 January 2023 from a realistic state of the atmosphere, leaving 2 days of spin-up which is sufficient given the size of our domain. We checked (not shown) that the uncertainty due to the start-up date is negligible compared to the uncertainty induced by the parameters we evaluate in the following.

One of the aims of the paper is to investigate the impact of horizontal resolution on simulated katabatic winds. Therefore, three different grids have been configured with horizontal resolutions of 10 km, 20 km chosen as the reference, and 40 km. Those grids are shown in Fig. 4, where the positions of the evaluation sites as red dots are in the center of the domain, far from the transition zone at the boundaries. As the katabatic forcing is proportional to the sine of the slope (Van den Broeke and Van Lipzig, 2003), the wind event is expected to be dependent on resolution through the discretization of the slope. We have updated the topography of LMDZ using the state-of-the-art dataset from Schaffer et al. (2016), which provides a more detailed and accurate map of the polar caps terrain elevation at a 1-km resolution. This high-resolution topography is then interpolated over the model grid resolution, and yields the terrain elevation shown in Fig. 4.

The dynamics timesteps are respectively 15 s, 30 s and 60 s for the 10-km, 20-km and 40-km resolution simulations. Unlike for global configurations, the physics timestep is set to 5 minutes instead of the standard 15 minutes due to stability issues that will be discussed in Sect. 4. Model outputs are then 10-minute averaged to match the AMRDC-AWS timestep.

It is important to note that at 40 km, the D18 cell is 31% oceanic. In LMDZ, each cell is divided into four tiles: land, ocean, sea ice and land ice. We only analyze the turbulence variables of the land ice tile, but the oceanic part of the cell may affect the mesh-averaged wind. For the other sites and resolutions, the cell is composed uniquely of "land ice". Model outputs are interpolated over a rectangular area around the exact localization of each station in order to avoid selection biases when a station is located near the border between two cells.

Wind speed at the first model level, typically between 8 and 10 m, is brought back to the height of the wind observations (6.7 m for D18 and 3 m for D47) assuming a logarithmic profile extrapolation using the model roughness length. Although

Figure 4. Terrain elevation of the 10-km, 20-km and 40-km grids. The D18, D47 and D85 are shown in red dots.



wind measurements are also carried at 2 m at the D18 station, we use the 6.7 m measurements to minimize the impact of the logarithmic extrapolation. To assess whether the local near-surface wind profile is logarithmic, we use the CALVA-D17 tower (see Barral et al. (2014)) 1 km downstream of D18 which comprises 6 wind speed levels (instead of 2 for the AWACA-D18 tower at the time of the case study). Wind measurements are plotted against log(z) for 14 30-minute profiles during the wind peak (between 04:00 and 10:00 LT). Corresponding linear regressions yielded a mean determination coefficient R<sup>2</sup> of 0.86. We therefore assume that a logarithmic approximation of the wind profile is reasonable.

Turbulent kinetic energy is linearly interpolated between its surface boundary condition, where it reads  $e = c_{\epsilon}^{2/3} u_{*}^{2}$  (Vignon et al., 2024), and its value at the first layer upper interface (TKE is defined at model layers' interfaces) of the model.

We then quantitatively investigate the parametric sensitivity of our results with perturbed parameter ensembles (PPE), i.e. a parametric exploration in the form of generalized sensitivity experiments without reference points. First, we carefully select a reduced number of key parameters amongst those we deem the most critical to simulate the katabatic layer, with realistic ranges of variation. Then we perform a latin hypercube sampling of those parameters, and we run ten times the number of parameters simulations to adequately explore the parametric uncertainty of the model. Thus, before looking at the observations, we are not able to favor a priori one configuration over the other. In this paper, we define the parametric uncertainty as the width of the envelope of the PPE.

The important issue to address is consequently: which parameters should we choose and what should be their ranges?

Turbulence has an essential role in shaping the katabatic boundary layer. Vignon et al. (2024) performed a parametric exploration using the history matching with iterative refocusing method (Williamson et al., 2013) and showed that for the new TKE-l turbulence parameterization, two parameters mostly determine the representation of weakly stable boundary layers such

as katabatic boundary layers:  $c_l$ , which controls the value of the mixing length in stratified conditions, and  $c_{\epsilon}$ , which modulates the amplitude of the diffusion coefficient. Vignon et al. (2024) further show that the uncertainty ranges are roughly [1; 2] for  $c_l$ , and [5; 10] for  $c_{\epsilon}$ .

Moreover, the surface drag exerted on the katabatic flow is driven by the momentum flux between the air and the snow cover (Eq. 5). We therefore add roughness length for momentum in the key parameters along with a parameter controlling the ratio between roughness length for heat and momentum  $r_{\frac{h}{m}}$ . We thus take into account the fact that  $z_{0h}$  is generally smaller than  $z_{0m}$  (van Tiggelen et al., 2023). Amory et al. (2017) explored the surface roughness at D17, one kilometer downstream of D18, and found seasonal values of the neutral drag coefficient at 10 m  $C_{DN10}$  corresponding to values of  $z_{0m}$  around  $10^{-3}$  m in January-February. Vignon et al. (2017a) found values of  $z_{0m}$  ranging from  $10^{-4}$  m to  $10^{-2}$  m on the Dome C plateau, except for very low temperatures never reached at D18 or D47. Hence we sample the interval  $[10^{-4}; 10^{-2} \text{ m}]$  for  $z_{0m}$ . Considering that  $z_{0m}$  varies over several orders of magnitude, this parameter is sampled logarithmically. We set the interval [0.1; 1] for the ratio  $r_{\underline{h}}$ , following the main variation range of this parameter found by Vignon et al. (2017b) at Dome C.

Then, we include parameters having an indirect impact on wind, e.g. by changing the near-surface temperature, stability and thus the katabatic forcing. Snow albedo and thermal inertia, both set constant in the model, are particularly essential in the control of surface temperature in Antarctica (Vignon et al., 2017b). We use the formula  $I = \sqrt{\lambda C}$  with  $\lambda$  the snow thermal conductivity estimated from Calonne et al. (2011); and C the snow specific heat estimated from Boone (2002), both parameters depending on snow density. Agosta et al. (2012) shows a rather narrow range of snow density values between 400 and 500 kg m<sup>-3</sup> over the region of interest; but to take lower densities due to fresh snow into account we lower the threshold to 250 kg m<sup>-3</sup> and come out with an interval of [250; 500] kg m<sup>-3</sup> for snow density, that yields an interval of [300; 800] J m<sup>-2</sup> K<sup>-1</sup> s<sup>-1/2</sup> for snow thermal inertia. Pristine snow albedo is mostly sensitive to grain size and solar zenith angle, and the variability is larger in the near-infrared domain (Grenfell et al., 1994). As the AWACA-D18 station reported temperatures near melting on the 18th of January that may have smoothed and widened snow grains, thus decreasing the albedo, we sample the large interval [0.4; 0.7] for snow near-infrared albedo.

Table 1 shows the six parameters retained as the most important ones. For each exploration, we run 10 times the number of parameters simulations, i.e. 60 simulations per exploration. In addition to the parametric uncertainty drawn by the PPE, we will look in depth at the following metrics: the mean near-surface wind and temperature during the peak, i.e. between 04:00 and 10:00 LT the 21 January. In the following section we present the results of the first exploration, i.e. the default configuration at 20 km.

#### 350 3 Results





#### 3.1 Analysis of the 20-km parametric ensemble

We begin the model evaluation and an in-depth analysis of the parametric exploration by the ensemble at 20-km resolution. The results of this exploration are presented in Fig. 5, where we compare the PPE to the AWACA-D18 and AWS-D47 observations;

| Symbol            | Description                                         | Range                                                                 |
|-------------------|-----------------------------------------------------|-----------------------------------------------------------------------|
| $c_l$             | controls the mixing length $l_s$                    | [1;2]                                                                 |
| $c_{\epsilon}$    | modulates the diffusion coefficient $\mathcal{K}_m$ | [5; 10]                                                               |
| $z_{0m}$          | roughness length for momentum                       | $[10^{-4};10^{-2}]\mathrm{m}$                                         |
| $r_{\frac{h}{m}}$ | ratio between $z_{0h}$ and $z_{0m}$                 | [0.1;1]                                                               |
| I                 | snow thermal inertia                                | $[300;800] \mathrm{J}\mathrm{m}^{-2}\mathrm{K}^{-1}\mathrm{s}^{-1/2}$ |
| $Ab_{ m NIR}$     | snow near-infrared albedo                           | [0.4; 0.7]                                                            |

Table 1. Parameters of the generalized sensitivity experiment, along with their uncertainty ranges. See text for details.


to LMDZ-6A which has the same physics but with the previous turbulent diffusion scheme used in CMIP6; and lastly to ERA5 reanalyses.

Fig. 5.a,d shows that wind speed observations range within the model parametric uncertainty in gray, indicating that ICOLMDZ is able to simulate correct wind speeds for this case study. In the same panels, although the model peak is flatter than the observations, the timing of the wind maximum during local early morning is well reproduced by the PPE. Parametric uncertainty, i.e. the width of the envelope drawn by the PPE, is wide and reaches approximately 6 m s<sup>-1</sup> at D18 and is slightly lower at D47 with 5 m s<sup>-1</sup>, representing approximately 35% of the maximum observed wind speed in both cases.

Fig. 5.b shows there is an underestimation of the turning of the wind just before (02:00) and after (09:00) the wind peak where the simulation ensemble does not overlap D18 observations. This is associated with a too smooth decrease in wind speed after the peak. Still in panel (b), unrealistic changes of wind direction are also produced by several simulations at D18 at 20:00 on the 20/01, and are associated with the lowest wind speeds (not shown). These overly weak winds correspond to shallow surface layers, and therefore to a too strong Ekman effect right above the surface. Conversely, all the simulations including LMDZ6-A and ERA5 commonly agree on the mean wind direction at D47 but this is on average 10° westward of the AWS-D47 reports. However, an offset problem in wind direction measurements cannot be excluded (David Mikolajczyk, personal communication) and wind direction observations at D47 are discarded in the following.

Concerning surface wind stress, the evolution of the model friction velocity is in fair agreement with the observations (Fig. 5.g), although the amplitude of the diurnal cycle is overestimated. The evolution of the simulated TKE in panel (h), which is very similar to wind speed and friction velocity, does not correspond to sonic anemometer observations until the wind has thoroughly decreased from 14:00, 21st January.

The observed TKE has a rather complex evolution throughout the event with values around  $2.5 \text{ m}^2 \text{ s}^{-2}$  alternating with peaks above  $3.5 \text{ m}^2 \text{ s}^{-2}$  before a gradual decrease together with wind speed. In-depth investigation of turbulence spectra during TKE peaks reveals a significant contribution of low-frequency eddies. The nature and source of those eddies remain to be explained but they are possibly associated with rotor-type eddies that form within a katabatic jump (Vignon et al., 2020). In fact, ICOLMDZ simulations suggest that the flow separation occurs near the D18 location and that the jump moves downstream as the wind accelerates - hence a reduction of the observed turbulence despite an increase in wind speed - and retreats northward

**Figure 5.** Top: Wind speed (a), Wind direction (b) and Temperature (c) at D18. Middle: Wind speed (d), Wind direction (e) and Temperature (f) at D47. Bottom: Friction velocity (g) and TKE (h) at D18. The PPE of the 20-km configuration is represented in gray; LMDZ-6A is in blue; ERA5 is shown for indication in red and observations are in black. Note that TKE observations in D18 are dotted where the number of non-valid data exceeds a threshold of 1%. An estimate of the observations uncertainties may be found in Fig. 2. Time is local time (UTC + 10).

during the end of the katabatic event, as is shown in the video supplement. This might explain why the observed TKE does not correlate with wind speed at least during the first part of the event. It is further worth noting that the local TKE-l turbulence

scheme of LMDZ is not well suited to simulate large eddies that form within the jump, and thus the TKE values and evolution associated with it. Testing the effect of a non-local mixing length such as that of Bougeault and Lacarrere (1989) and Rodier et al. (2017) is an avenue for further research but beyond the scope of the present work.

Note that due to the presence of moderate drifting snow, sonic anemometer data with more than 1% of non-valid data are dotted to indicate a lower degree of confidence. Nonetheless, we checked that filling non-valid data with random values taken in the 30-minute averaging intervals has a small impact on turbulence outputs (less than 10%, not shown). Therefore, even if we cannot quantify the impact of drifting snow on valid data, this simple test increased our level of confidence in turbulence measurements and contributed to not discard them even during the wind peak. Thus, there seems to be a structural difference in the model TKE compared to the observations.







Still in Fig. 5.h, oscillations of TKE are clearly visible in LMDZ-6A simulation with the previous turbulent diffusion scheme. These oscillations are replicated on the LMDZ-6A wind speed in Fig. 5.a and 5.d. They will be discussed in Sect. 4. It is worth noting that the ERA5 wind speed (10-meter wind speed brought back to the observation height with a logarithmic extrapolation using  $z_{0m}$ =0.03 m, the roughness length used in ERA5 to derive near-surface winds over polar caps) is quite different from the other datasets at D18 (Fig. 5.a). This discrepancy might be explained by an unrealistic roughness length for this region, a too coarse resolution to represent the complex topography near the coast, or the fact that only the AMRDC-AWS are assimilated by the reanalyses (though in general, only pressure and temperature), and not the AWACA-D18 mast. It also confirms that the ERA5 nudging above the boundary layer exerts a limited constraint on the model near-surface winds.

While temperature observations at D18 and D47 (Figs. 5.c and 5.f) range within the PPE during daytime, the model evolution is not sharp enough and observations come out of the parametric uncertainty during local night. At 20 km resolution, the D18 grid cell altitude is 462 m compared to 468 m in reality; and the D47 grid cell altitude is 1533 m compared to 1560 m in reality. These differences are too small and highly unlikely to explain the discrepancies between the model temperatures and the observations. This structural deficiency of the model questions the representation of surface snow (density, albedo) and its thermal properties.

When looking at wind profiles (Fig. 6a,c), we see that the spread of the PPE is mostly pronounced in the first 300 m, and that the shape of the profiles does not vary much. There is nonetheless a difference in the jet nose altitude: weaker surface wind speeds peak higher in altitude and were found to be associated with more mixed katabatic layers. The wind direction profiles in Fig. 6b,d exhibit a backing pattern above the jet nose altitude due to the decreasing friction away from the surface (Renfrew and Anderson, 2006).

Vignon et al. (2019) show using 8 years of radiosondes that the observed katabatic layer at Dumont d'Urville is much less peaked and shallow than the D18 ICOLMDZ wind profile (Fig. 6.a). Of course, this comparison is limited as our case study may not be representative of semi-climatological observations, and as the vertical structure of the air flow above DDU may be impacted by the presence of the archipelago islands, the aftermath of a katabatic jump, and shallow convection over open ocean. In the model, the wind profile above DDU (not shown) is however quite similar to D18, although slightly less peaked and shallow.

**Figure 6.** Wind speed and direction vertical profiles at D18 (a,b) and D47 (c,d) for the 20-km PPE averaged during the wind speed peak (04:00-10:00).

It is worth noting that a similar PPE exercise was carried out using a standard buoyancy length scale formulation for the mixing length with no dependency on the wind shear (Eq. 2). No major differences in the results have been observed neither on the surface wind speed nor on the vertical profile, suggesting that the feedback of the wind shear upon the mixing length is not sufficiently strong in our case study to modify the overall vertical mixing in the katabatic layer.

Overall, we see that although the parametric uncertainty of the model is quite large, all simulations exhibit the same general evolution. In the following, we explore which parameters determine the near-surface wind and temperature spread.

## 3.2 Parametric sensitivity of the 20-km parametric ensemble






For the 20-kilometer resolution configuration, we investigate which parameters are the most critical for the control of wind speed and temperature magnitudes, and thus explain the ensemble dispersion. For this purpose, we consider 2 metrics: the averages of wind speed and temperature during the maximum intensity of the event between 04:00 and 10:00 LT the 21/01. We plot these metrics in Figs. 7 and 8 as functions of the six parameters of the exploration for the 60 simulations. For each subplot, a linear regression is computed along with a 95 % confidence interval to assess whether there is a statistically significant correlation.

Fig. 7 shows that the most critical parameter for the wind speed during the peak is the roughness length  $z_{0m}$ , as it is the only variable exhibiting a statistically significant correlation with wind speed for both D18 and D47, except for snow near-infrared albedo at D18 (although much less clearly). As expected, slower winds correspond to larger roughness lengths, i.e. to stronger surface drags. The slight positive correlation between wind speed and  $Ab_{\rm NIR}$  at D18 likely indicates a decrease of the katabatic forcing via an increase of near-surface temperature. The turbulent diffusion parameters  $c_l$  and  $c_{\epsilon}$  that control the vertical transport of momentum do not have any first-order effect on the near-surface wind speed magnitude during the case study. These two parameters induce more mixing but their effect is of 2nd order compared to  $z_{0m}$ .

The linear regression of the mean horizontal wind speed during the peak on the logarithm of  $z_{0m}$  at D18 and D47 shows that the critical value of the model roughness length to match the observed wind speed is 0.51 mm at D18 and 0.46 mm at D47, assuming that the measurement heights are respectively 6.7 m and 3 m. Thus, the roughness length value required to match the wind at D18 is only slightly different from the one required to match D47 observations. However, this comparison is very sensitive to the uncertainty on the measurement height of the wind observations at D47, and reliable instrument height records would have been crucial to draw robust conclusions on this aspect.

Although the impact of snow near-infrared albedo  $Ab_{\rm NIR}$  on wind speed magnitude is limited compared to  $z_{0m}$ , Fig. 8 shows that it becomes the most critical parameter for explaining near-surface temperature both at D18 and D47. Observations as black dotted line outside the cloud points confirm the fact that model near-surface temperatures do not decrease enough during the night. Neither roughness length nor the other parameters related to heat fluxes, i.e.  $r_{\frac{h}{m}}$  or snow thermal inertia I, exhibit a statistically significant correlation with near-surface temperatures, and therefore do not have any first-order impact. Choosing the maximum value  $Ab_{\rm NIR} = 0.7$  to minimize the near-surface temperature positive bias during the night is not a proper solution as it increases the negative bias during the day. The dependency of albedo to the size of snow grains or the solar zenital angle is not represented in LMDZ, despite the critical role of the albedo on near-surface temperature. Further work is needed on this aspect to correctly evaluate the model bias. Moreover, this study focuses on a clear-sky event, and conclusions can be different with more complex cloudy conditions. However, all the interpretations presented above also hold for the other two resolutions that will be explored in the next section (not shown).

The same analysis is then conducted on the model friction velocity  $(u_*)$  and TKE (e) and reveals more contrasted results (Figs. not shown). Beyond the quite expected positive correlation between  $u_*$  and  $z_{0m}$  through the Monin-Obukhov logarithmic approximation of near-surface wind (see Eq. 7), and between e,  $c_\epsilon$  and  $z_{0m}$  through the linear interpolation of the TKE between

Parameter

Figure 7. Mean near-surface wind speed between 04:00 and 10:00 LT the 21/01 as a function of all the parameters for the 60 simulations of the 20-km resolution PPE (see Table 1), with corresponding determination coefficients  $R^2$ .

its surface boundary condition  $e = c_{\epsilon}^{2/3} u_*^2$  and the first layer upper interface of the model, both variables also exhibit a fairly good correlation with  $Ab_{\rm NIR}$ , indicating again the dependency of wind on albedo. Thus, no parameter emerges as the most

Parameter

Figure 8. Same as Fig. 7 but for near-surface temperature.

critical for the control of the turbulence variables, and further investigation beyond the scope of this study needs to be carried out to unravel the properties of the turbulence's parametric sensitivity.

# 3.3 Sensitivity of the simulations to horizontal resolution

After having investigated the impact of the different parameters on the PPE, we now dig into the impact of horizontal resolution. As a reminder, katabatic winds are expected to be resolution-dependent particularly because the katabatic forcing term depends on the slope (Van den Broeke and Van Lipzig, 2003) which is more or less properly resolved depending on the horizontal grid spacing. The same set of parameters is used for each configuration, differences come only from the resolution. This makes it possible to disentangle to what extent the choice of the resolution is controlling the representation of the wind with respect to the parametric sensitivity.

**Figure 9.** Wind speed at 6.7 m at D18 (a) and at 3 m at D47 (b), for the 10-km (blue), 20-km (green) and 40-km (red) resolutions. Colored lines represent the median of each PPE, with the corresponding envelope in shading. Observations are in black.

We plot in Fig. 9 the simulation ensemble for the wind speed at the 3 resolutions (10, 20 and 40 km) at the D18 and D47 sites, along with the observations. It shows that the 10-km and 20-km resolution ensembles exhibit very similar results, whereas the 40-km configuration is different during the wind peak. Although the 40-km simulation ensemble is much less accurate at D18 (i.e., farther from the observations), it is also more precise (i.e., with a smaller variance). Student's t-tests over the ensembles of 60 simulations show that the mean wind speed during the peak (04:00 to 10:00) of the 40-km PPE statistically differs from the two others (p-values 

Figure 10. Wind speed over the Adélie Land slope averaged between  $138^{\circ}$ E and  $140^{\circ}$  E and during the wind maximum between 04:00 and 10:00 LT the 21/01 for the 3 resolutions. Potential temperature is shown in contours, and the position of D18 and D47 in black dotted lines. The same set of parameters is used for the 3 resolutions with  $z_{0m} = 0.5$  mm and  $Ab_{NIR} = 0.64$ .

Concerning other variables such as friction velocity and TKE at D18, the conclusions are similar: the 10-km and the 20-km PPE are alike, whereas the 40-km ensemble differs. On the vertical, the katabatic jet nose is sharper and shallower for the 10-km and 20-km resolution than for the 40-km resolution configuration and for the two sites, although the differences are small (not shown). The three configurations exhibit the same relation between the surface wind speed and the wind peak altitude, i.e. stronger wind speeds peak lower in altitude.



We conclude that for our case study, results are similar for a 10-km and 20-km horizontal resolution. Conversely, the 40-km resolution appears too coarse to properly reproduce the key processes at the coast. This is consistent with Vignon et al. (2019) who conclude that a 27 km resolution of the Polar-WRF model is sufficient for modeling the wind over the slope of Adélie Land, but not right at the coastal edge where the katabatic jump extends too far offshore. A resolution of 20 km seems an upper-bound to correctly capture the transition of katabatic winds at the coast.

# 4 Discussion on critical but underappreciated aspects of Antarctic katabatic flow modeling

In the last section, we investigated which part of the model bias was associated with parameter calibration, with horizontal resolution and with structural deficiencies in the model. Before concluding, we deem important to discuss three particular aspects we encountered in the study, which although critical, are often underappreciated in the modeling of katabatic winds.

## 4.1 Parameterization of roughness length over Antarctic snow surfaces






In Sect. 3.2, we show that the roughness lengths required to match observations at D18 and D47 are slightly different, and that a fixed  $z_{0m}$  over all the continent may not be sufficient to represent the katabatic flow along the slope. In the following, we implement as an illustration a parameterization of roughness length in ICOLMDZ, which we evaluate with  $z_{0m}$  observations derived from the sonic anemometer.

A straightforward approach is to differentiate the D18 and D47 sites with temperature (mean temperatures during the case study respectively at D18 and D47: around -10 °C and -20 °C). Indeed the sastrugi, which represent the better part of surface roughness in this region of Antarctica, adapt more rapidly to the mean wind when the temperature is lower (Amory et al., 2017) because of less efficient snow sintering. Conversely, warmer snow is more cohesive and sastrugi take more time to adjust to the mean wind, inducing a larger  $z_{0m}$ . This reasoning is consistent with the larger  $z_{0m}$  found for D18, the warmer of the two sites. Amory et al. (2017) developed an empirical parameterization of a temperature-dependent roughness length for Antarctica. This parameterization is particularly relevant for our case study as it has been derived from observations at the D17 site, only one kilometer downstream of D18. In order to test this parameterization in ICOLMDZ, we compute a roughness length from the D18 observations using the sonic anemometer output  $u_*$  and the formula:

$$z_{0m} = H\exp(\frac{-\kappa U}{u_*})\tag{7}$$

with H the measurement height (here H = 2 m),  $\kappa$  the von Karman constant, U the wind speed and  $u_*$  the friction velocity. This formula is valid only for neutral and stationary conditions, and we apply the same data selection criteria as in Amory et al. (2017).

Fig. 11 shows that the observed  $z_{0m}$  at D18 is around 0.5 mm during the wind peak, which is fairly similar to the model  $z_{0m}$  required to match wind speed observations found in Sect. 3.2 but is only half the default constant value (1 mm) in the model.

On the contrary, the roughness length obtained with the temperature-dependent parameterization (red line in Fig. 11) is between 2 and 3 times larger than the observations, and follows unrealistically the diurnal cycle of temperature which is even in phase opposition with the observed  $z_{0m}$ . Therefore, we apply a low-frequency filter to the temperature to smooth out the diurnal cycle, but the parameterized  $z_{0m}$  in brown line in Fig. 11 is still several times larger than the observations and does not reproduce a realistic evolution of roughness length. Indeed, Amory's parameterization is fitted over several years of data, and the temperature dependency in the parameterization firstly reflects the monthly to seasonal variability.

We therefore conclude that this parameterization is not adapted to our case study, and that a more physical parameterization of  $z_{0m}$  is needed in the model. Moreover, even if the model  $z_{0m}$  needed to match wind observations at D18 is in good agreement

**Figure 11.** Roughness length of the model using the parameterization of Amory et al. (2017) depending on the instantaneous temperature (red line) or the 2-day smoothed temperature (brown line) at the D18 site. Observations are shown in black dots when the stability and stationarity conditions criteria are met.

with the observed  $z_{0m}$ , comparing a model roughness length representative of a whole cell (20 km) with an observed and very local roughness length that may vary widely over a few kilometers is delicate, and conclusions should be considered with caution.

# 4.2 Oscillations in katabatic flows



The LMDZ-6A simulation exhibits oscillations in the wind speed timeseries (see Fig. 5.a,d). In fact, katabatic flows are prone to natural and numerical oscillations. As an in-depth investigation of our simulations reveals that all of them - not only LMDZ-6A - exhibit oscillatory patterns, a discussion on this aspect is needed.

McNider (1982) shows that oscillations in katabatic winds may occur under stable stratification by the combination of buoyancy and adiabatic effects. As the air goes down Antarctic slopes, it is adiabatically warmed and decelerates due to its lower density. This slower air parcel undergoes radiative cooling that increases its density, and is then accelerated again. These oscillations have been successfully reproduced numerically by Chemel et al. (2009) with a period of 10-11 minutes. No such oscillations have been found on power density spectra of the original 1-min data of the AWACA-D18 mast in January 2023. We expect this phenomenon to be more visible in winter when radiative cooling is stronger. The 10-min sampling time of the AMRDC-AWS observations does not allow us to check for the presence of these oscillations at D47.

Even though "natural" oscillations are not very probable in our case study, the strong wind speed and the coupling between wind shear and turbulent diffusion in such flows make our simulations prone to artificial numerical oscillations.

The first type of oscillations was encountered only in the LMDZ-6A configuration, which is the CMIP6 version of LMDZ with the previous version of the TKE-l scheme by Yamada (1983). Blue line in Fig. 5.h shows spurious oscillations of the TKE so-called fibrillations, which are due to the coupling between wind shear and vertical turbulent diffusion (Girard and Delage, 1990). These fibrillations increase when the physics timestep, i.e. the timestep of the physical parameterizations, is larger. They create small oscillations of the wind speed (blue line in Fig. 5.a,d), although the average wind speed remains correct. These oscillations are strongly reduced when using the new turbulent diffusion scheme of Vignon et al. (2024), which uses a more robust numerical integration of the TKE equation with in particular a full implicit treatment of the dissipation term.

The second type of oscillations appears in the wind speed timeseries when using the standard physics timestep of 15 minutes instead of the 5 minutes we used so far. With such a setup, katabatic air masses have time to cross an entire cell without experiencing the surface drag. Indeed, with wind speeds representative of our case study ( $\approx 15 \text{ m s}^{-1}$ ) the distance covered by an air parcel during one physics timestep approaches the size of the grid cell, especially for the 10-km configuration. This CFL (Courant-Friedrichs-Lewy condition) violation-type issue creates numerical modes visible as striped patterns (see Fig. 12.d). These oscillations, quite visible with a physics timestep of 15 minutes (Fig. 12.c) in the 10-km resolution configuration, are attenuated with a timestep of 5 minutes (Fig. 12.b), and completely disappear with a timestep of 1 minute (Fig. 12.a). Obviously, a trade-off must be found between reducing these oscillations as much as possible while not prescribing a too short and numerically costly physics timestep. Thus, a physics timestep of 5 minutes seems to be a reasonable tradeoff.

The last type of oscillations is hidden at the scale of the dynamics timestep (respectively 15, 30 and 60 s for the 10-km, 20-km and 40-km resolutions). When plotting the timeseries of the wind speed at the dynamics timestep (instead of the physics timestep resolution so far) in Fig. 13, it exhibits a non-realistic sawtooth pattern, an issue already raised in Krinner (1997) almost thirty years ago. This is due to the fact that the surface drag and turbulent diffusion is applied in one shot at every physics timestep. Krinner (1997) proposed to "distribute" physics tendencies at each dynamics timestep, which gives a smoother curve in Fig. 13. However, these tendencies are computed explicitly at the beginning of the physics timestep and do not take into account the evolution of the air mass within the timestep. As such an option is not appropriate for other parameterizations such as deep convection, this calls for further work on the physics-dynamics in ICOLMDZ.

It is worth mentioning that the second type of oscillations mentioned above can still be found in the smooth simulation (not shown), indicating that these 2 oscillations are of a different nature and have to be resolved separately.

## 4.3 Subgrid orographic drag







In Caton Harrison et al. (2024), the major current issues about modeling coastal near-surface winds in Antarctica are listed and discussed. One of the key points amongst those exposed is how wind drag due to subgrid-scale orography is handled in climate models. In ICOLMDZ, the parameterization of subgrid-scale orography (SSO) on the atmospheric flow is based on the works of Lott and Miller (1997) and Lott (1999). It produces first, a gravity wave drag due to wave breaking in the middle atmosphere and to the dissipation and breaking of trapped lee waves (Lott, 1998) and second, a low-level drag and lift forces at the model levels that intersect the SSO. The low-level drag force represents the blocking effect of orography leading to a flow separation at the relief flanks and it is opposed to the local wind. The lift represents the effect of blocked air in narrow valleys intensifying

**Figure 12.** Wind speed at D47 using a 1-min (a), 5-min (b) and 15-min (c) physics timestep for the 10-km configuration at the first model level. Below (d), an example of the wind oscillation patterns that propagate along the slope for a physics timestep of 15 minutes the 21 January at 20:15.

Figure 13. Wind speed around the peak at the dynamics timestep, with (red) or without (blue) the smoothed physics tendencies option.

the vortex compression. The intensity and direction of the forces depend of SSO properties such as the SSO standard deviation, mean slope and anisotropy, which are computed from a reference high-resolution topography dataset (see Appendix of Lott and Miller (1997) for details). The parameterization itself is based on a conceptual model of flow-topography interaction which considers a representative mountain with an elliptical shape. This paradigm is reasonable for typical mountain ranges or hilly terrains, but it is not adapted for the typical relief found over ice sheet plateaus which consist in gently sloping terrains - with slight convexity or concavity - towards the ocean. Over such terrains, even though there are no mountains the SSO standard deviation is non-null due to the mean slope. Hence, the parameterization activates leading to unrealistic drag and lift effects on the flow, and particularly on the low-level katabatic winds. Such spurious effects were already noticed in Pithan et al. (2015) (see their Appendix A), who consequently chose to deactivate the lift part of the scheme over glaciers. For the coarser 40-km configuration, we find that the contribution of subgrid orographic drag to the sum of the absolute values of all the wind tendencies is 7% at D18 and 24% at D47, at the first level of the model and averaged during all the period of the case study. Despite this moderate impact at 40-km that becomes negligible at higher resolutions, we expect this issue to be critical in global simulations with lower resolutions such as the IPSL-CMIP7 runs (of resolution  $\approx 150$  km) for which the subgrid-scale contribution to the grid cell topography becomes substantial.

To avoid spurious drags on the Antarctic katabatic flow due to an inappropriate activation of the ICOLMDZ SSO scheme on ice sheets, we propose a new criterion such that the parameterization activates only where the number of subgrid-scale mountains  $N_{\rm SSO}$  is greater than 0. The definition of  $N_{\rm SSO}$  is the number of triangular mountains in a mesh for a given mean subgrid-scale slope and standard deviation, and it reads :

$$N_{SSO} = \frac{\zeta_{SSO}\Delta x}{4\sigma_{SSO}} - 1 \tag{8}$$

where  $\zeta_{\rm SSO}$  is the SSO mean slope,  $\sigma_{\rm SSO}$  the SSO standard deviation and  $\Delta x$  the mesh size. Applying this criterion makes it possible to keep the activation of the SSO parameterization everywhere on the continents except on the ice sheet plateaus and escarpment regions (see Fig. 14). However, the criterion does not deactivate the parameterization over the coastal margins of Antarctica and Greenland, regions with no mountains but where the steepness of the terrain is particularly abrupt and where orographic gravity waves can be excited (Watanabe et al., 2006; Alexander and Murphy, 2015; Vignon et al., 2020). This overall invites to further work on model development to make the SSO drag parameterization adapted to this specific relief type.

# 5 Conclusions

Katabatic winds are a key feature of the climate of Antarctica, that impact the continental scale as well as the global scale through the creation of coastal polynyas connecting with sea-ice dynamics and oceanic overturning circulation. However, biases persist in atmospheric models whose origins sometimes remain ambiguous amongst interactions with synoptic conditions, horizontal resolution - in relation with topography effects particularly at the coast where an abrupt transition occurs, and parameterization of turbulent mixing in the boundary layer and surface fluxes as well as their parameters' calibration.

**Figure 14.** Grid cells where the subgrid orographic drag is deactivated (blue) or retained (yellow) for an ICOLMDZ simulation with a resolution of 150 km similar to CMIP7. Oceanic points are blanked.

The aim of this study is to investigate the physical behavior of katabatic winds in the LMDZ atmospheric model, and to disentangle which part of the biases can be explained by a poor parameter calibration, a too coarse resolution or structural deficiencies in the model physics. The LMDZ model is coupled to the new dynamical core DYNAMICO in a limited area configuration to reach higher horizontal resolution and smaller computing cost. A katabatic-driven wind event is carefully selected in January 2023 in clear-sky conditions, and a regional configuration of ICOLMDZ is set up over the Adélie Land region with 3 spatial resolutions of 10, 20 and 40 km. ICOLMDZ is evaluated through perturbed parameter ensembles (PPE) that explore the role of 6 key parameters within realistic intervals of variation. Simulations are compared to in-situ observations at the D18 and D47 sites, leveraging recent and original turbulence measurements at D18.



The parametric exploration of the 20-km resolution ensemble shows that the model is able to represent the katabatic wind event as the observations range within the parametric uncertainty. Parametric uncertainty reaches approximately 35% of the maximum wind intensity. The evolution of the turbulent kinetic energy (TKE) differs significantly between ICOLMDZ and the sonic anemometer measurements during the first part of the event, which might be attributed to a poor representation of the dynamics of the katabatic jump. The variation of near-surface temperature is reasonably reproduced by the model except a warm bias during the core of the night. Vertical wind speed profiles seem little impacted by the parametric exploration. We find that weaker near-surface winds peak higher, and vice versa. However, further wind profile observations are needed to correctly evaluate the modeled katabatic layer vertical structure in Adélie Land, and particularly at the margin of the ice sheet.

An analysis of the impact of the parameters shows that roughness length  $(z_{0m})$  is the most critical parameter explaining the magnitude of the wind peak and the spread within the ensemble, whereas near-infrared snow albedo is the most critical parameter for the magnitude of the near-surface temperature during the wind maximum. The values of  $z_{0m}$  required to match wind observations slightly differ for D18 and D47, but this comparison is quite sensitive to the uncertainty in the instrument height

at D47.

The parametric exploration is then conducted over three horizontal resolutions (10, 20 and 40 km). The 10-km configuration does not bring a significant improvement compared to the 20-km configuration in terms of surface wind because of a too wide parametric uncertainty. On the contrary, the 40-km configuration differs from the other resolutions, and fails to properly capture the behavior of the katabatic flow closer to the coast.

We then complement the analysis with a discussion on three critical but underappreciated aspects of katabatic flow modeling: the parameterization of roughness length over ice sheets, "natural" and numerical oscillations in katabatic flows that constrained the physics timestep to five minutes, and the representation of the drag due to subgrid-scale orography in Antarctica.




As roughness length is the most critical parameter for the magnitude of the wind speed during the case study, one key message that arises from this study is that the development of a physical parameterization of roughness length in ICOLMDZ is needed to properly simulate katabatic winds all along the slopes of Antarctica. Structural biases in near-surface temperatures indicate that the representation of the ICOLMDZ snow cover needs to be refined, with a particular focus on snow albedo. Further investigation could dig into the role of the dynamics-physics coupling, as well as the parameterization of the subgrid-scale orographic drag in coastal Antarctica.

The role of large-scale forcing on katabatic winds, and to what extent synoptic conditions are well represented by a global configuration of the model, remain to be explored. Another lead is to investigate the interactions between blowing snow and katabatic winds through the development of a parameterization of blowing snow in ICOLMDZ (ongoing work, see Vignon et al. (2025)). All these questions will benefit from the observations of the AWACA project that are currently deployed at D17, D47, D85 and Dome C, and in particular from the meteorological masts measuring wind and turbulence. These autonomous sites will operate all year long, hopefully providing precious data in winter when more extreme katabatic winds develop.

Code and data availability. Data from the AWACA-D18 mast during the case study are available here: https://web.lmd.jussieu.fr/~vwiener/data\_for\_review/. They will be made available on Pangaea (https://www.pangaea.de/) once this paper receives a DOI. The eddy covariance code used to derive turbulence estimations from sonic anemometer data is available here: https://gitlab.in2p3.fr/valentin.wiener/processing-code-sonic. Data from the AMRDC-AWS can be accessed here: https://doi.org/10.48567/4ed8-bq91. ERA5 data are available on the Copernicus Climate Data Store website here: https://cds.climate.copernicus.eu/datasets/reanalysis-era5-single-levels?tab=download. The LMDZ and DY-NAMICO models are freely distributed at the following links https://web.lmd.jussieu.fr/~lmdz/pub/ and https://gitlab.in2p3.fr/ipsl/projets/dynamico/dynamico. The version used for the specific simulation runs for this paper is the "svn" release 4950 from 22 May 2024.

*Video supplement.* An animation of the katabatic jump as represented by ICOLMDZ for the 10-km resolution simulation is available here: https://web.lmd.jussieu.fr/~vwiener/data\_for\_review/. It will be made available on the German National Library of Science and Technology (TIB) once this paper receives a DOI.

Author contributions. VW, EV, CG and AB designed the study. VW, FT and CG designed the AWACA-D18 mast and deployed it on the field. VW ran the simulations, VW and EV performed the analysis and VW wrote the first draft of the manuscript with inputs from EV.
 YM helped setting up the model configuration. TCH performed the budget decomposition analysis to ascertain the katabaticness of the case study. GCR helped VW to code the processing algorithm to retrieve turbulence outputs from the D18 sonic anemometer. VW, EV, TCH, CG, FT and AB discussed and revised the manuscript.

Competing interests. The authors declare that they have no conflict of interest.

Acknowledgements. This work has been financed by the AWACA project, ERC synergy grant n° 951596. The field campaign and instruments deployment have made been possible with the financial and logistical support of the French Polar Institute IPEV, program 1251. Simulations were performed using HPC resources from the IDRIS (Institut du Développement et des Ressources en Informatique Scientifique, CNRS, France), project RLMD, allocation number n° AD010107632R1. The analysis benefited from the work of the DEPHY research group, funded by CNRS/INSU and Météo-France. We also gratefully thank David Mikolajczyk and Matthew Lazarra for their help with the AWS dataset, François Lott for fruitful discussions about the parameterization of SSO in coastal Antarctica, Charles Amory for discussions about the z<sub>0m</sub>,
 Lea Raillard for having paved the way towards a working configuration of the limited-area ICOLMDZ model, Maëlle Coulon–Decorzens for her help to setup and understand the parametric explorations, and Alice Maison for helpful discussions about near-surface turbulence. We also thank Jacques Couzinier for his help in the development and validation of the sonic anemometer processing algorithm. Finally, we thank Cécile Agosta for invaluable feedbacks on the manuscript.

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
