# Peer review of "An extensive investigation of the ability of the ICOLMDZ model to simulate a katabatic wind event in Antarctica."

_EGUsphere, 2025_

## Author Comment (AC1)

**Response to the reviews**

Firstly, the authors thank the 2 reviewers for having carefully read our work, and prompted valuable suggestions that will improve the manuscript. We answer below (in blue) to the reviewers' comments (in black).

**Response to RC1:**

**Major comments:**

My major criticism of the approach in this paper is that the vertical structure of the wind is not adequately discussed, even though the authors use tower data to validate their model. Specifically, the "roughness length" is a parameter that tunes the vertical profile of the wind, but the hypothesis that a logarithmic vertical profile of the wind is valid is never discussed or demonstrated.

Indeed, this aspect was not sufficiently discussed in the manuscript.

**Added in Sect. 2.4:**

"To assess whether the near-surface local wind profile is logarithmic, we use the CALVA-D17 tower (see [2]) 1 km downstream of D18 which comprises 6 wind speed levels (instead of 2 for the AWACA-D18 tower at the time of the case study). Wind measurements are plotted against log(z) for 14 30-minute profiles during the wind peak (between 04:00 and 10:00 LT). Corresponding linear regressions yielded a mean determination coefficient R2 of 0.86. We therefore assume that a logarithmic approximation of the wind profile is reasonable."

The data and measurements altitudes of the CALVA-D17 meteo mast can be accessed here: https://web.lmd.jussieu.fr/~cgenthon/SiteCALVA/Datas/d17-vent23.dat.

Moreover, when computing roughness length z0 in Sect. 4.1 from sonic anemometer data, we use the stability and stationarity criteria proposed in Sect. 4.1 of Amory et al 2017 [1] to assess neutral conditions during which the logarithmic shape of the wind profiles is generally respected. The stationarity criterion is valid 55% of the time, including during the wind peak (black dots on Fig. 11), whereas the stability criterion associated to a logarithmic profile is always valid.

The vertical wind shear is very important for vertical momentum and heat transport, and it should be within the scope of the paper to discuss this aspect also. (It is mentioned briefly starting line 296, but never evaluated).

The vertical wind shear is only available for the AWACA-D18 meteo mast. Indeed, the AWS-D47 mast only comprises one measurement level, and no observations cover at least the first 2 layers of the model (first level at  $\sim$ 8-10 m). Concerning the AWACA-D18 observations, the wind shear  $\Delta u/\Delta z$  follows the evolution of wind speed and reaches a maximum of 0.57 m/s/m at 07:00 LT the 21/01, i.e. at the same time as the wind speed peak (see Fig. 1).

As the TKE does not follow the same pattern, it hints that turbulence does not originate from local wind shear generation, but rather from lower frequencies processes as mentioned in Sect. 3.1.

Added in Sect. 2.2 : "The wind shear between 2 and 7 m follows the same evolution and simultaneously reaches a maximum of  $0.6 \text{ m s}^{-1}\$  (not shown)."

Figure 1: Wind shear at D18 between 2 and 7 m.

The wind profiles of the ensemble are shown on Figure 6, but not evaluated against observations. I understand you may not have observations for this particular event, but other climatological validation must be available, or a more detailed critical examination of the literature and its known limitations is needed.

Climatological observations of wind profiles are indeed available at Dumont d'Urville (DDU) thanks to daily radiosoundings. Fig. 3 below extracted from Vignon et al, 2019 [4] shows that the observed katabatic layer is much less peaked and shallow than that of the ICOLMDZ model visible in Fig. 6 of the present manuscript. Besides, our case study may not be representative of climatological observations. However, this bias cannot be simply explained by different simulated vertical structures of the katabatic flow above DDU and D18. Indeed, the wind profile above DDU (see Fig. 2) is quite similar to D18, although slightly less peaked and shallow. In reality, the air flow above DDU may be impacted by the presence of the archipelago islands, the aftermath of a katabatic jump, as well as shallow convection over open ocean. Moreover, radiosoundings are only launched during local mid-morning, i.e. not during the end of the night when the katabatic forcing is strongest. As mentioned in the manuscript, this is the reason why we did not include the 21/01/2023 radiosounding in our study, as it was launched several hours after the wind maximum. To our knowledge, no other observations of wind profiles are available in the vicinity of D18. Further work needs to be done to correctly evaluate modeled wind profiles in coastal Adélie Land, that could benefit from the AWACA wind profiler newly deployed at Dumont d'Urville, when the data will be available.

Figure 2: The PPE wind speed profiles at DDU (left) and D18 (right) for the 20-km configuration.

**Added this in Sect. 3.1:**

"\cite{Vignon\_2019} show using 8 years of radiosondes that the observed katabatic layer at Dumont d'Urville is much less peaked and shallow than the D18 ICOLMDZ wind profile (Fig. \ref{vertical\_profile}.a). Of course, this comparison is limited as our case study may not be representative of semi-climatological observations, and as the vertical structure of the air flow above DDU may be impacted by the presence of the archipelago islands, the aftermath of katabatic jumps, and shallow convection over open ocean. In the model, the wind profile above DDU (not shown) is however quite similar to D18, although slightly less peaked and shallow."

**Added this in the Conclusion:**

"However, further wind profile observations are needed to correctly evaluate the modeled katabatic layer vertical structure in Adélie Land, particularly at the margin of the ice sheet."

**Figure 2.** Vertical profiles of the annual wind speed (top row), temperature (middle row) and relative humidity with respect to ice (bottom row) from radiosonde measurements at nine Antarctic stations. Black lines are the medians, colored lines refer to the 10th, 20th, 30th, 40th, 60th, 70th, 80th and 90th percentiles. In the legend, "Pctx" refers to the shaded area that covers x percent of the data greater than the median and x percent of the data lower than it. The altitude z is above ground level. Numbers in exponential form next to station names in the title indicate the number of radiosoundings per day at the corresponding station. The "\*" symbol labels the two stations for which only data from December to February are shown.

Figure 3: See legend above. Extracted from Vignon et al, 2019 [4].

As an illustration, Fig. 4 shows the PPE of the 3 configurations during the 21/01/2023 radiosounding around 10:00 LT. This figure underlines the need to better constrain simulated wind profiles with new observations, as neither the calibration nor the horizontal resolution can explain the differences with the radiosoundings. This difference not only in shape but also in the integrated momentum hints that surface roughness is locally increased by the presence of the archipelago islands, which are not represented at the horizontal resolution of the ICOLMDZ model.

Figure 4: Wind speed (left) and direction (right) PPE profiles at DDU on the 21/01/2023 10:00 LT for the 10 (blue), 20 (green) and 40 (red) kilometer configurations. Radiosoundings observations are in black.

Also, in section 4.1, you show the limits of the existing roughness length parametrisations, and how using z0 to tune the wind results in over-tuning other deficiencies in representing near surface wind. Since the roughness length is the critical parameter you have identified, it need to be explained and assessed a bit better.

Clarifications have been added in the text (see the LaTeX diff file).

Here are a few line-by-line comments:

Line 345: space before ";" to be removed. -> Done.

Figure 9: To ease the comparison between the various resolutions, it would be better tp put a,b and c in the same graph, or at least to have the range of the other simulations put on each graph. You can use transparency to put all the simulations together, or you could simplify your display by showing a swath between the min and max of your parametric ensemble, that could be overlayed between the 3 resolutions. —> Done (see Fig. 5).

Figure 12: put a, b, and c in the same plot, it will save you a lot of space.

We agree that it would save a lot of space, but the oscillation patterns are much less visible when the three curves are superimposed. After consideration, we have made the choice to keep the initial Figure.

Figure 5: Wind speed at 6.7 m at D18 (a) and at 3 m at D47 (b), for the 10-km (blue), 20-km (green) and 40-km (red) resolutions. Colored lines represent the median of each PPE, with the corresponding envelope in shading. Observations are in black.

**Response to RC2:**

**Major comments**

Introduction: when you discuss the impact of model horizontal resolution, an interesting citation is: doi:10.3189/2012JoG12J020.

-> done. Added the sentence : \citet{Lenaerts\_2012} assess with a regional climate model that enhancing horizontal resolution (from 27 to 5.5 km) strongly reduces wind speed biases in Adélie Land.

Fig. 2: Given that katabatic forcing depends on absorbed solar radiation, hence solar incidence angle, would it not be more logical to use local time?

You are right. UTC time has been changed to local throughout the figures and the text in the manuscript.

1. 198: Fig. 3 shows that large-scale forcing is still significant; would it not have been more logical to use this parameter as the first selection threshold (e.g. >80%)?

Large-scale forcing is not negligible but overall relatively weak (< 30%) for the D18 and D47 sites, formerly indicated by red crosses in Fig. 3. We increased the crosses size and changed their color to make them more visible.

1. 251: It surprises me that z0 and zh are chosen the same, while many studies (including over glaciated surfaces) imply zh << z0, see e.g. Fig. 4 in doi:10.1029/2022JD036970.

While z0 = zh in the LMDZ default configuration, we introduced in our parametric study the ratio between z0h and  $z0m \ll r_h/m$  » (see line 4 of Table 1) ranging between 0,1 and 1. We find no significant correlation between this parameter and the wind speed magnitude (see Fig. 7,8).

To clarify this point, we added in Sect. 2.4 : "We thus take into account the fact that  $z_{0h}$  is generally smaller than  $z_{0m}$  \citep{Van\_Tiggelen\_2023}."

Fig. 5: Is there a way to find out whether the oscillations in wind speed come from the turbulent exchange or the other way around?

This is a quite delicate point and it is in fact a chicken-and-egg problem. The wind shear and the turbulent diffusion are tightly coupled through the TKE shear production term. The oscillation (fibrillation) can thus emerge owing to this coupling, especially for long time steps which push the convergence and stability of the numerical integrations of the TKE equation and turbulent diffusion to their limits. Even though much care was taken for the numerical treatment of the turbulent diffusion and TKE equations, the katabatic flow with very strong wind shear and TKE remains challenging for models with a typical time step of a few minutes.

Please define the phrase "katabatic jump". If it simply refers to the transition from a katabatic (over the ice sheet slopes) to a non-katabatic regime (e.g. a situation with cold air piling up over the flat ice shelves, sea ice or ocean), such a jump must occur simply because the slope vanishes. What is the relation to 'hydraulic jumps', i.e. the transition from supercritical (Fr>1) to subcritical (Fr<1) flow?

The katabatic jump is indeed due to the slope break, but also corresponds to a transition regime in the air flow from supercritical to subcritical (see Yu et al 2007 [6]). Hence the parallel with the hydraulic jump such as made in Gallée and Pettré, 1998 [3].

Added in the text: From supercritical (Froude number \$Fr\$>1), the flow turns subcritical (\$Fr\$<1) \citep{Yu\_2007}. This phenomenon, which often occurs along the coast of Antarctica, is hereafter referred to as a katabatic jump (e.g., \citet{Gallee 1998, Vignon 2020}).

Fig. 10: This figure shows near-zero values some distance above the surface, which I cannot reconcile with Fig. 6.

Near-zero values (<2 m/s) in Fig. 10 occur downstream of D18 for the default 20-km configuration. At the D18 location, wind speed values in altitude fall mostly in the 4-6 m/s bin, which is consistent with the ensemble in Fig. 6. We also thickened the black dotted lines indicating D18 and D47 locations.

l. 471: Effect of horizontal resolution on katabatic winds: what is the influence of 3D topographic features (e.g. channelling, convexity/concavity) on this dependency?

As mentioned in Sect. 3.3 of the MS, Yu et al 2007 [6] show that the katabatic flow is influenced both by the steepness and the variation of the underlying slope. Using a non-hydrostatic model, they simulated a sudden cessation of wind speed near the foot of the slope for a concave underlying slope such as in the vicinity of D18.

To assess whether the dependency to convexity is affected by horizontal resolution, a coarser topography could be applied to the 10-km configuration while keeping the same mesh. This could be done in future work on the subject.

**Minor comments**

1. 21: 'than the air above'. It is the density contrast with the air at the same elevation but away from the surface, in combination with the surface slope, that sets up the katabatic pressure gradient force.

Changed to: This near-surface air, colder and therefore denser than the air at the same elevation but further downslope, ...

1. 24: 'and may exceed 30 ms-1.' This is an arbitrary number; they may also exceed 40 ms-1. Please make it more specific (by mentioning average wind speeds and including a citation).

Changed to : and may exceed 15 m s\$^{-1}\$ in yearly average in certain regions such as Adélie Land \citep{Parish 1988}.

- 1. 25: solar radiation -> surface absorbed solar radiation -> Done.
- 1. 89: Greenland ice cap -> Greenland ice sheet -> Done.
- 1. 100: "collected in the Adélie Land steep slope." Please reformulate.

collected in coastal Adélie Land.

l. 117: When you provide values for 'near-surface winds', please also provide the measurement height or the height they were corrected to. -> Added.

1.120: "slightly sloped". This contrasts with the statement in 1. 110 "the slope is steep near the coast". I think the layman would not consider these slopes 'steep'.

The slope is steeper near the coast, [...], and becomes flatter as we approach the Plateau.

1. 120: no -> neither

neither blue ice nor outcropping rocks

l. 123: "are characterised by small roughness lengths but that may spread over very variable scales." Can you make this statement more concrete by providing some examples? This could also be done in the introduction.

Added : For example,  $\citet{Vignon_2017b}$  find roughness length values between  $10^{-5}$  and  $10^{-2}$  m at Concordia.

1. 159: Cloud cover impacts katabatic forcing not directly but through the surface energy balance. Suggest removing here.

We agree. Changed to: "as well as cloud cover and blowing snow particles that can **both** affect the surface energy balance"

- 1. 173: Remove 'and' -> Done.
- 1. 176: ultrasound -> ultrasound -> Done.
- 1. 201-204: Surely the accelerations cannot be calculated so accurately. Suggest to remove the last digit (i.e. 9.07 m s-1 h-1 -> 9.1 m s-1 h-1). -> You are right. Done.

- 1. 229: The wind shear expression appears wrong. -> We use the same formulation for the vertical wind shear as in Eq. 8 of Vignon et al, 2024 [6].
- 1. 272: nudged by -> nudged to -> Done.
- 1. 292, 293: "land ice". Why the quotation marks? -> Quotation marks removed.
- 1. 303 and throughout manuscript: Please refrain from using qualifications such as "thoroughly" or "extensively" when referring to your investigation. Same with 1. 304: "carefully", 1. 307 "adequately". Better to add quantitative metrics when applicable.
- 1. 303 : thoroughly -> quantitatively.
- 1. 311: "history matching exploration", please clarify.

Changed to : \citet{Vignon\_2024} performed a parametric exploration using the history matching with iterative refocusing method \citep{Williamson 2013} and showed that ...

- 1. 348: winds -> wind speeds -> Done.
- 1. 355: narrow -> shallow (?) -> Done.
- 1. 367: within katabatic jump -> within a katabatic jump -> Done.
- 1. 377: I would qualify <10% as 'small' rather than 'negligible'. -> Done.
- 1. 459: This "looser" katabatic flow", please reformulate.

This more extended katabatic flow ...

- 1. 485: colder -> lower (please check throughout MS) -> Done.
- 1. 486: a warmer snow -> warmer snow -> Done.
- 1. 514: lighter density -> smaller density -> Done.
- 1. 525: Explain 'physics timesteps'.

Changed to: These fibrillations increase when the physics timestep, i.e. the timestep of the physical parameterizations, is larger.

- 1. 526: " although the magnitude of the wind remains correct ". If the oscillations are numerical, the wind speed is no longer correct. Consider: "average wind speed". -> Done.
- 1. 585: subsist -> persist (?) -> Done.
- 1. 602: conversely -> vice versa (?) -> Done.
- 1. 610: fail -> fails -> Done.
- 1. 612: remove 'work' -> Done.
- 1. 613: ice caps -> ice sheets -> Done.

**References**

- [1] Amory, C., Gallée, H., Naaim-Bouvet, F., Favier, V., Vignon, É., Picard, G., Trouvilliez, A., Piard, L., Genthon, C., and Bellot, H.: Seasonal variations in drag coefficient over a sastrugicovered snowfield in coastal East Antarctica, Boundary-Layer Meteorology, 164, 107–133, 2017.
- [2] Barral, H., Genthon, C., Trouvilliez, A., Brun, C., & Amory, C.: Blowing snow in coastal Adélie Land, Antarctica: three atmospheric-moisture issues, The Cryosphere, 8(5), 1905-1919, 2014.
- [3] Gallée, H. and Pettré, P.: Dynamical constraints on katabatic wind cessation in Adélie Land, Antarctica, Journal of the atmospheric sciences, 55, 1755–1770, 1998.
- [4] Vignon, É., Traullé, O., and Berne, A.: On the fine vertical structure of the low troposphere over the coastal margins of East Antarctica, Atmospheric Chemistry and Physics, 19, 4659–4683, 2019.
- [5] Vignon, É., Arjdal, K., Cheruy, F., Coulon-Decorzens, M., Dehondt, C., Dubos, T., Fromang, S., Hourdin, F., Lange, L., Raillard, L., et al.: Designing a fully-tunable and versatile TKE-l turbulence parameterization for the simulation of stable boundary layers, Journal of Advances in Modeling Earth Systems, 16, e2024MS004 400, 2024.
- [6] Yu, Y., Cai, X., and Qie, X.: Influence of topography and large-scale forcing on the occurrence of katabatic flow jumps in Antarctica: Idealized simulations, Advances in Atmospheric Sciences, 24, 819–832, 2007.